# Long Term Results of Single-Fraction Carbon-Ion Radiotherapy for Non-small Cell Lung Cancer

**DOI:** 10.3390/cancers13010112

**Published:** 2020-12-31

**Authors:** Takashi Ono, Naoyoshi Yamamoto, Akihiro Nomoto, Mio Nakajima, Yuka Isozaki, Goro Kasuya, Hitoshi Ishikawa, Kenji Nemoto, Hiroshi Tsuji

**Affiliations:** 1Department of Radiation Oncology, QST Hospital, 4-9-1 Anagawa, Inage-ku, Chiba-shi, Chiba 263-8555, Japan; yamamoto.naoyoshi@qst.go.jp (N.Y.); nomoto.akihiro@qst.go.jp (A.N.); nakajima.mio@qst.go.jp (M.N.); isozaki.yuka@qst.go.jp (Y.I.); kasuya.goro@qst.go.jp (G.K.); ishikawa.hitoshi@qst.go.jp (H.I.); tsuji.hiroshi@qst.go.jp (H.T.); 2Department of Radiation Oncology, Faculty of Medicine, Yamagata University, 2-2-2, Iida-Nishi, Yamagata 990-9585, Japan; knemoto@ymail.plala.or.jp

**Keywords:** heavy ion radiotherapy, dose fractionation, radiation, carcinoma, non-small cell lung

## Abstract

**Simple Summary:**

There were no reports on long-term results of single-fraction passive carbon-ion radiotherapy in patients with early-stage non-small cell lung cancer. We showed that this treatment was not inferior compared to stereotactic body radiotherapy or proton beam therapy with no ≥grade 2 pneumonitis. This study suggests that single-fraction passive carbon-ion radiotherapy can serve as an alternate treatment for patients with early-stage non-small cell lung cancer, especially in medically inoperable patients.

**Abstract:**

Background: The purpose of the present study was to evaluate the efficacy and safety of single-fraction carbon-ion radiotherapy (CIRT) in patients with non-small cell lung cancer. Methods: Patients with histologically confirmed non-small cell lung cancer, stage T1-2N0M0, and treated with single-fraction CIRT (50Gy (relative biological effectiveness)) between June 2011 and April 2016 were identified in our database and retrospectively analyzed. Toxicity was evaluated using the Common Terminology Criteria for Adverse Events version 4.0. Results: The study included 57 patients, 22 (38.6%) of whom had inoperable cancer. The median age was 75 years (range: 42–94 years), and the median follow-up time was 61 months (range: 6–97 months). The 3- and 5-year overall survival rates were 91.2% and 81.7%, respectively. All survivors were followed up for more than three years. The 3- and 5-year local control rates were 96.4% and 91.8%, respectively. No case of ≥ grade 2 pneumonitis was recorded. Conclusions: This study suggests that single-fraction CIRT for T1-2N0M0 non-small cell lung cancer patients is feasible and can be considered as one of the treatment choices, especially in medically inoperable patients.

## 1. Introduction

Lung cancer remains the leading cause of cancer incidence and mortality globally, with 2.1 million new lung cancer cases and 1.8 million deaths predicted in 2018, which accounts for 18.4% of all cancer deaths [1]. Even in stage I lung cancer patients, the 5-year overall survival (OS) rate without any treatments is 4.0–10.1% [2,3].

In early-stage lung cancer, surgery is the mainstay [4], but not all patients can undergo surgery. Stereotactic body radiotherapy (SBRT) is one of the treatment choices for these patients [5,6,7], and published meta-analyses show that SBRT is as effective as surgery in terms of OS [8,9].

At our institution, we have been treating lung cancer patients with carbon-ion radiotherapy (CIRT) since 1994 [10]. It has the advantage of reducing the irradiation dose to normal tissue due to its increased energy deposition, with a penetration depth up to a sharp maximum at the end of its range, also known as the Bragg peak [11]. We conducted a dose-escalation trial to determine the optimal dose fractionation schedule in 2011 [12], and we have been treating lung cancer patients using 50Gy (relative biological effectiveness (RBE)) in a single fraction. In May 2016, the dose delivery method changed from passive scattering to pencil beam scanning. Therefore, the objective of this study was to evaluate long-term clinical results of 50Gy (RBE) in a single fraction using passive CIRT.

## 2. Results

### 2.1. Patients

We identified 125 lung cancer patients who received single-fraction passive CIRT (50Gy (RBE)) between June 2011 and April 2016. Sixty-eight patients were excluded owing to interstitial pneumonitis (*n* = 18), lack of histological confirmation (*n* = 46), non-Japanese ethnicity (*n* = 2), uncontrolled esophageal cancer (*n* = 1), or concomitant lung cancer (*n* = 1). Eventually, 57 patients met the inclusion criteria, including 22 (38.6%) medically inoperable patients (Table 1). Treatment was successfully administered to all patients. There were no patients who received adjuvant chemotherapy after CIRT. The median diameter of lung cancer was 26.0 mm (range: 11.0–46.0 mm). The median follow-up time was 61 months (range: 6–97 months), and all survivors were followed up for more than 3 years. Fifty-five patients’ volume–dose histogram data were available. The mean value of mean lung dose (MLD) was 2.64Gy (RBE).

### 2.2. Survival

The 1-, 3-, and 5-year OS rates were 96.5% (95% confidence interval (CI): 91.8–100%), 91.2% (95% CI: 83.9–98.5%), and 81.9% (95% CI: 71.7–92.1%), respectively (Figure 1). Five patients died from lung cancer progression and 11 patients died of diseases other than lung cancer—other cancers after CIRT (*n* = 2), cardiovascular diseases (*n* = 4), respiratory diseases (*n* = 2), digestive diseases (*n* = 1), liver failure (*n* = 1), and renal dysfunction (*n* = 1). The 3- and 5-year OS rates for patients in stage IA were 89.7% (78.6% for inoperable and 96.0% for operable patients), and 81.8% (71.4% for inoperable and 88.0% for operable patients), respectively, whereas the corresponding rates for those in stage IB were 94.4% (87.5% for inoperable and 100% for operable patients) and 82.1% (75.0% for inoperable and 87.5% for operable patients), respectively. In multivariate analyses, operability and sex were regarded as significant factors for OS (Table 2).

### 2.3. Local Control

Four patients had local recurrence within the irradiated field at 14–51 months after CIRT. The 1-, 3-, and 5-year local control (LC) rates were 100%, 96.4% (95% CI: 91.5–100%) and 91.8% (95% CI: 84.0–99.6%), respectively (Figure 2). One of the four patients had local recurrence with pleural dissemination and received chemotherapy. The remaining three patients received salvage local treatments, one patient underwent salvage surgery, and two patients received salvage CIRT with 72Gy (RBE) in 12 fractions. The 3-year LC rates of stage IA and IB were 97.4% (95% CI: 92.3–100%) and 94.1% (95% CI: 83.0–100%), respectively. There were no significant predictive factors for LC in the univariate analysis, and we did not perform multivariate analysis for LC (Table 3). Although there was no significant difference in LC between patients with stages IA and IB, stage IB patients had a local recurrence (three of 18 patients) more frequently than stage IA patients (one of 39 patients). Furthermore, two of three patients with stage IB had local recurrence ≥4 years after CIRT.

### 2.4. Toxicities

There were no ≥ grade 3 toxicities (Table 4). Four patients (7.0%) had grade 2 rib fractures and two (3.5%) had peripheral motor neuropathy. All 4 patients who had symptomatic rib fractures were relieved by using analgesics. On the other hand, both patients with peripheral motor neuropathy received CIRT for non-small cell lung cancer (NSCLC) of the lung apex. They had mild symptoms without requiring any treatment.

## 3. Discussion

To the best of our knowledge, this is the first study of long-term clinical results of 50Gy (RBE) in single-fraction CIRT in the world.

CIRT for stage I NSCLC seems promising considering the results of previous Japanese studies of SBRT and proton beam therapy (PBT) for NSCLC, although the patients’ backgrounds was not the same (Table 5) [7,13]. Nagata et al. reported that OS was much better for operable than for inoperable patients [7]. The OS was better for operable patients in our study too. Although more than 30% of the patients in our study were inoperable, the OS was much better than that of operable patients who received SBRT. A possible reason for this is that our study was retrospective, but all survivors were followed up for ≥3 years. In addition, male sex was a significant factor for poor OS in our study. A meta-analysis by Nakamura et al. support our data, but the conclusive reason why female patients with NSCLC live significantly longer remains unclear [14].

Regarding surgery for lung cancer, Okami et al. reported the results of a large population study of surgery cases with lung cancer [15]. They found that the 5-year OS rates of patients with stages IA and IB were 88.9% and 76.7%, respectively. The OS of patients who received CIRT was not inferior to that of patients who had surgery. Single-fraction CIRT may become a treatment choice for both operable and inoperable patients, although attention should be given to local recurrence.

Local recurrence is one of the biggest challenges in treating lung cancer patients with radiotherapy, and careful long-term follow-up examinations are necessary to assess treatment outcomes correctly. In our study, local recurrence was observed in approximately 10% of the patients within 5 years, and in two cases, more than 4 years after CIRT. Some patients with local recurrence after radiotherapy are eligible for salvage surgery [16,17] or reirradiation [18,19]. Dickhoff et al. reported that salvage surgery for local recurrence after SBRT was technically feasible with acceptable 90-day mortality (0–11%) in their systematic review [16]. Mizobuchi et al. also reported results of salvage surgery after CIRT. Among 95 patients who underwent CIRT and then experienced local recurrence, only 12 patients were medically operable. Surgery was feasible with a good OS (3-year OS was 82% after salvage surgery) without any life-threatening complications [17]. These findings suggest that salvage surgery is a good treatment option for local recurrence after CIRT. However, we must provide a sufficient explanation for primary treatment methods because most of the patients should receive extended resection compared to the surgery before CIRT [17]. In the present study, one of four patients with local recurrence 4 years after CIRT successfully underwent salvage surgery and survived for 1 year without further recurrence. Two patients with local recurrences received re-CIRT, and one had no recurrence for 42 months, although eventually the patient had local recurrence and pleural dissemination. Another patient had local recurrence 8 months after re-CIRT. Karube et al. reported results of re-CIRT for 29 patients with local recurrence, and the 2-year OS and LC rates were 69% and 66.9%, respectively, without any ≥ grade 3 late toxicities, although grade 2 late pulmonary toxicities were observed in 6.9% of the patients [18]. Although there have been no reports on photon reirradiation therapy for local recurrence after CIRT, Viani et al. published a meta-analysis of re-SBRT for local recurrence after first SBRT [19]. They reported that the estimated 2-year OS and LC were 54% and 73%, respectively, but the estimate of ≥grade 3 toxicities was 9.8%. The OS and LC following re-CIRT were similar to that of re-SBRT with low toxicity, and re-CIRT may become a treatment of choice for local recurrence after CIRT. However, the 2-year LC of re-CIRT was much lower, and grade 2 late pulmonary morbidities were higher than that of primary CIRT in a previous study. Moreover, the OS was lower than that of patients who received salvage surgery. One of two patients who received re-CIRT had local recurrence and pleural dissemination 42 months after reirradiation. Another patient had local recurrence 8 months after re-CIRT. Although salvage surgery may be the first-line treatment method, various salvage treatments, including re-CIRT, may become one of the treatment choices for local recurrence after CIRT.

Previously, we reported the efficacy of CIRT (50Gy (RBE) in a single fraction), and it was expected that the LC rate for patients with T2a would be higher than 80% [12]. In the present study, the 5-year LC rate of patients with T2a NSCLC was higher, but this rate does not seem to be sufficient. Chen et al. reported that pathological examinations revealed that there were viable cells in the center of 3 of 5 local recurrent tumors (four local recurrences were T1 lung cancer, and one was T2) [20]. Thus, total irradiation doses were probably insufficient to control tumors, especially in T2 lung cancer, and further dose escalation may be one of the choices to improve the outcomes. Since May 2016, patients have received pencil beam scanning CIRT for lung cancer at our institution. This technique modulates the radiological dose. Therefore, increasing the dose to the center of the tumor without increasing the peripheral dose may be one of the options for decreasing local recurrence. 

Radiation pneumonitis is also a serious complication of lung cancer radiotherapy. There was no ≥ grade 2 radiation pneumonitis in the present CIRT study. According to previous studies with four fractions as well as in our study, ≥ grade 3 pneumonitis after CIRT was less frequent than after SBRT or PBT [7,13,21,22]. Even compared to single-fraction SBRT reported in the Radiation Therapy Oncology Group 0915 [23], single-fraction CIRT had fewer ≥ grade 2 respiratory toxicities (0% in the present study vs. 13% in single-fraction SBRT). A possible explanation is that CIRT reduces the surrounding lung dose compared to SBRT or PBT [24,25]. Regarding irradiation doses and volumes to the lung in SBRT, Barriger et al. reported that the development of symphonic pneumonitis correlated with the MLD and lung volumes irradiated at least 20Gy [26]. Ebara et al. reported that CIRT reduced irradiated normal lung volume compared to SBRT, especially in low-dose areas [24]. In fact, the MLD of the present study was 2.64Gy (RBE), lower than that of SBRT (4.14Gy) [26]. Moreover, the concentration for target volume was higher than that for SBRT. Demizu et al. reported that CIRT reduces irradiated lung doses compared to PBT while maintaining equal coverage for lung tumors [25]. In fact, SBRT for early lung cancer is a good treatment option considering the OS and LC in a previous study [7,23], but CIRT may lead to low respiratory toxicities with good OS and LC due to the low irradiation volume compared to SBRT. In the present study, we did not find any factors for pneumonitis because there was no ≥grade 2 radiation pneumonitis. Further studies are needed.

Regarding other toxicities, there were some patients who had grade 2 peripheral motor neuropathy and rib fracture. If patients with lung cancer did not receive radical treatment, the OS time was short [2,3]. Moreover, symptoms of rib fracture can be controlled by analgesics and symptoms of peripheral motor neuropathy were mild. So, we think those toxicities were acceptable. However, we should explain those risks, especially when tumors were at lung apex, and reduce those toxicities as far as possible. To solve these problems, dose constraints for peripheral motor neuropathy are currently under consideration.

There were some limitations to the present study. The study was retrospective and performed at a single institution. Furthermore, the number of patients was relatively small. The number of CIRT fractions has decreased stepwise from 18 to 9, 4, down to a single fraction, at our institution [10,27,28]. Following a dose-escalation study with a single-fraction of 28Gy (RBE) initiated in April 2003, the optimal dose for stage I lung cancer was determined as 50Gy (RBE), which was the planned maximum dose in the dose-escalation study [12]. Thus, this is the first report on treatment outcomes of CIRT using the optimal dose in a single fraction for stage I NSCLC and 50Gy (RBE) has been further confirmed as the standard dose.

## 4. Materials and Methods

### 4.1. Patients

Patients treated with CIRT between June 2011 and April 2016 at the QST hospital were identified in our database and retrospectively analyzed. Patients’ cancer stages were determined according to the Union for International Cancer Control, 7th edition [29] and based on computed tomography (CT) and positron emission tomography (PET)-CT. The inclusion criteria were as follows: Japanese patients who had received 50Gy (RBE) in single-fraction passive CIRT; those with solitary lung tumors; those with pathologically confirmed NSCLC by bronchoscopy or needle biopsy; those with a World Health Organization performance status of 0–2; those having no lymph node metastasis; those with absence of distant other organ metastasis or other sites of uncontrolled cancer. Patients who had received concurrent chemotherapy, prior irradiation in the same area, and those with interstitial pneumonitis were excluded. In present study, there were no NSCLCs near the pulmonary hilum, because we treated those lung cancers by CIRT in 12 fractions instead of a single fraction.

### 4.2. Carbon-Ion Radiotherapy

Treatment planning for CIRT was based on three-dimensional CT images that were taken at 2.5 or 5.0 mm intervals in the exhalation phase. All patients were immobilized in the supine or prone position using Moldcare (Alcare, Tokyo, Japan) or Shelfitter (Kuraray, Osaka, Japan). CIRT planning was performed using the HIPLAN (National Institute for Radiological Science, Chiba, Japan), which was developed at our institution until July 2013, and Xio-N (ELECTA, Stockholm, Sweden; Mitsubishi Electric, Tokyo, Japan). Gross tumor volume (GTV) included the lung tumor. The clinical target volume (CTV) was defined as GTV plus a margin of 0.5–1.0 cm. Spicule formations and pleural indentations were included as appropriate. The planning target volume (PTV) was the CTV, plus the internal margin, which considered the respiratory movements of each patient’s tumor. The patients were rotated by a maximum of ± 20° with 3–4 different angle ports between the horizontal and vertical beams. Carbon-ion beams of 250, 350, or 400 MeV, depending on the target size, and water equivalent path length along the beamline, were generated by the heavy ion medical accelerator in Chiba. The total dose was applied to the isocenter and tuned to cover the PTV with a 95% isodose line of the prescribed dose. Figure 3 shows the clinical results in a patient who received this dosing scheme. The CIRT field was positioned with digitally reconstructed radiographs, X-ray imaging, and metal markers made of iridium wire as landmarks. The CIRT dose is shown in Gy (RBE), which was calculated by multiplying the physical dose by the RBE.

### 4.3. Evaluation and Follow-Up

Follow-up visits occurred in 1–3-month intervals the first year and every 3–6 months thereafter; almost all patients underwent a CT or PET-CT scan. Pneumonitis, rib fracture, pleural effusion, peripheral motor neuropathy, and dermatitis radiation were evaluated using the Common Terminology Criteria for Adverse Events version 4.0 [30].

### 4.4. Statistical Analysis

Statistical analyses were performed using the IBM SPSS Statistics software program (version 22, SPSS Inc., Chicago, IL, USA). OS was defined as the time between the day of CIRT and the time of the last follow-up examination or death. LC time was defined as the time between the day of CIRT and the time of the last follow-up or local recurrence. The Kaplan–Meier method was used to estimate the OS and LC probability. Univariate and multivariate Cox regression analyses were performed to investigate the risk factors for OS and LC. Significant tendency factors (*p* < 0.1) in the univariate analysis were included in the multivariate analysis. All p-values were two-sided and *p*-values of <0.05 were considered statistically significant. Chronic obstructive pulmonary disease was defined as forced expiratory volume in 1 s less than 70% of the predicted volume. 

## 5. Conclusions

This study suggests that CIRT in a single fraction is feasible and can be considered as one of the treatment choices for early lung cancer patients.

## Figures and Tables

**Figure 1 cancers-13-00112-f001:**
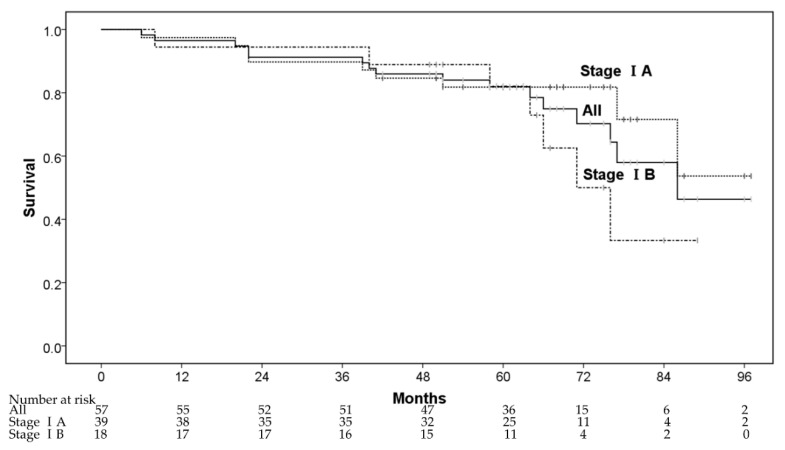
The overall survival rate.

**Figure 2 cancers-13-00112-f002:**
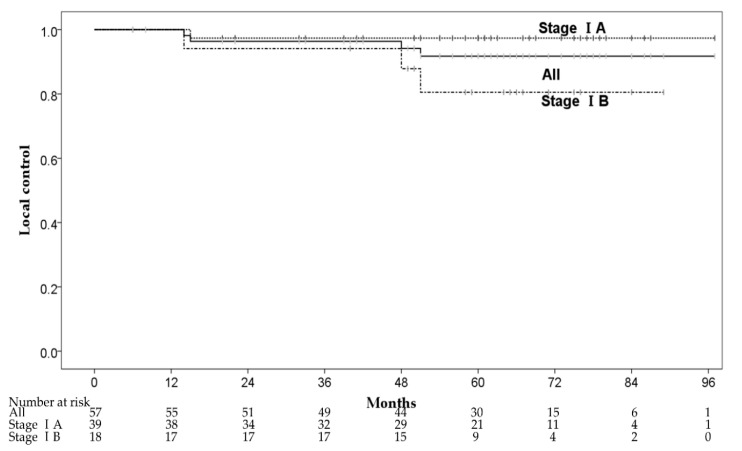
The local control rates.

**Figure 3 cancers-13-00112-f003:**
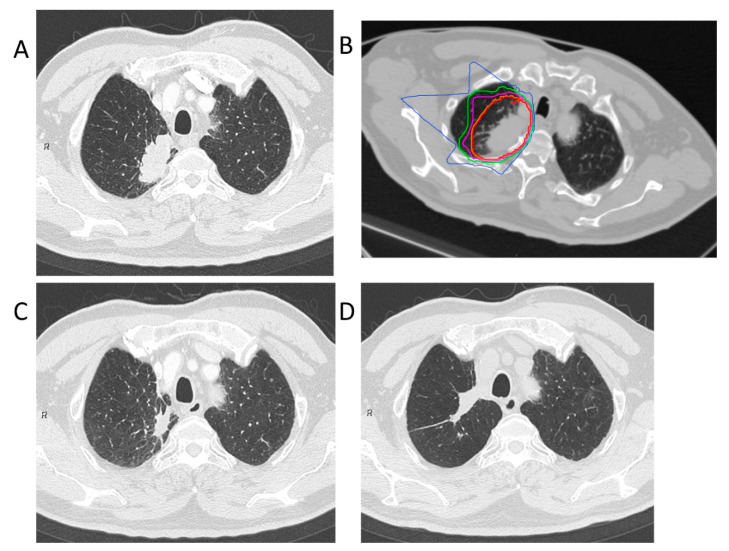
Carbon-ion radiotherapy for the right upper lobe adenocarcinoma (stage IB). (**A**) Computed tomography before treatment. (**B**) The dose distribution for lung cancer. The red, orange, purple, green, and blue lines showed dose levels of 95%, 90%, 70%, 50%, and 30%, respectively. (**C**) Computed tomography 3 months after treatment. (**D**) Computed tomography 84 months after treatment. This patient had no local recurrence.

**Table 1 cancers-13-00112-t001:** The patient characteristics (*n* = 57).

Characteristics	Patients
Age (years)	
Median (range)	75 (42–94)
Gender	
Male	32 (56.1%)
Female	25 (43.9%)
Performance status	
0	47 (82.5%)
1	9 (15.7%)
2	1 (1.8%)
Operable or inoperable	
Operable	35 (61.4%)
Inoperable	22 (38.6%)
Follow-up time (months)	
Median (range)	61 (6–97)
Stage (UICC 7th)	
IA	39 (68.4%)
IB	18 (31.6%)
Stage (UICC 8th)	
IA2	24 (42.1%)
IA3	15 (26.3%)
IB	13 (22.8%)
IIA	5 (8.8%)
Tumor location	
Right upper lobe	20 (35.1%)
Right lower lobe	10 (17.5%)
Left upper lobe	12 (21.1%)
Left lower lobe	15 (26.3%)
Histopathology	
Adenocarcinoma	38 (66.7%)
Squamous cell carcinoma	17 (29.8%)
Non-small cell lung cancer	2 (3.5%)
Chronic obstructive pulmonary disease	
Yes	22 (38.6%)
No	35 (61.4%)
Diameter of lung tumor (mm)	
Median (range)	26.0 (11.0–46.0)

**Table 2 cancers-13-00112-t002:** Univariate and multivariate analysis for overall survival.

Factor	Patient (*n* = 57)	5 Year OS	Univariate Analysis		Multivariate Analysis	
			HR (95% CI)	*p*-Value	HR (95% CI)	*p*-Value
Age			1.74 (0.63–4.77)	0.286	-	-
≤75	29	81.8%				
>75	28	82.1%				
Gender			2.80 (0.92–8.58)	0.071	3.35 (1.04–10.82)	0.043 *
women	25	91.8%				
men	32	74.0%				
Performance status			2.26 (0.71–7.15)	0.167	-	-
0	47	84.7%				
1–2	10	70.0%				
Operability			3.53 (1.21–10.34)	0.021 *	3.93 (1.33–11.59)	0.013 *
operable	35	88.1%				
inoperable	22	72.7%				
Histlogy			0.91 (0.33–2.54)	0.860	-	-
others	19	84.2%				
adenocarcinoma	38	81.1%				
Statge			1.80 (0.67–4.86)	0.248	-	-
IA	39	81.8%				
IB	18	82.1%				
COPD			1.38 (0.51–3.73)	0.528	-	-
no	35	79.2%				
yes	22	86.4%				

Abbreviations: OS: overall survival; HR: hazard ratio; CI: confidential interval; COPD: chronic obstructive pulmonary disease. * *p*-value < 0.05.

**Table 3 cancers-13-00112-t003:** Univariate analysis for local control.

Factor	Patients (*n* = 57)	5 Year LC	Univariate Analysis	
			HR (95% CI)	*p*-Value
Age			0.35 (0.035–3.28)	0.351
≤75	29	88.6%		
>75	28	94.7%		
Gender			3.04 (0.31–26.41)	0.337
women	25	95.5%		
men	32	88.9%		
Performance status			0.04 (0.00–6401.09)	0.596
0	47	90.4%		
1–2	10	100%		
Operability			0.70 (0.07–6.81)	0.760
operable	35	90.8%		
inoperable	22	95.0%		
Histlogy			1.50 (0.16–14.43)	0.726
others	19	93.8%		
adenocarcinoma	38	91.1%		
Statge			6.59 (0.68–63.39)	0.103
IA	39	97.4%		
IB	18	80.5%		
COPD			5.68 (0.59–54.71)	0.133
no	35	96.3%		
yes	22	84.4%		

Abbreviations: LC: local control; HR: hazard ratio; CI: confidential interval; COPD: chronic obstructive pulmonary disease.

**Table 4 cancers-13-00112-t004:** Toxicities.

Toxicities	Grade 0	Grade 1	Grade 2	Grades 3–5
Pneumonitis	0	57 (100%)	0	0
Rib fracture	23 (40.4%)	30 (52.6%)	4 (7.0%)	0
Pleural effusion	49 (86.0%)	8 (14.0%)	0	0
Peripheral motor neuropathy	55 (96.5%)	0	2 (3.5%)	0
Dermatitis radiation	12 (21.1%)	45 (78.9%)	0	0

**Table 5 cancers-13-00112-t005:** Comparison of clinical results of radiotherapy for non-small cell lung cancer in Japan.

Authors	Total Number	Stage	Number of Patients	Median Follow-Up Time	Treatment Methods	3-Year OS	5-Year OS	3-Year LC	Grade ≥3 Pneumonitis
Nagata et al. [7]	168	IA	104 (inoperable)	47 months	SBRT	59.9%	42.8%	87.3%	8.7%
			64 (operable)	67 months		76.5%	54.0%	85.4%	3.1%
Ohnishi et al. [13]	440	IA	277	38 months	PBT	80.7%	-	92.2%	1.0%
		IB	163			73.0%	-	79.0%	3.6%
Present study	57	IA	39	61 months	CIRT	89.7%	81.6%	97.4%	0
		IB	18			94.4%	82.1%	94.1%	0

Abbreviations: SBRT: stereotactic body radiotherapy; PBT: proton beam radiotherapy; CIRT: carbon-ion radiotherapy.

## Data Availability

Data sharing is not applicable to this article.

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
