# Peer review of "Long Term Results of Single-Fraction Carbon-Ion Radiotherapy for Non-small Cell Lung Cancer"

_cancers, 2020, doi:10.3390/cancers13010112_

Round 1

Reviewer 1 Report

General Comments

              This single-institution study was written well and can be published. I would like the authors to clarify one fact: There are 4 out of 57 patients that got grade 2 rib fractures for lung apex tumors. Do you still plan to treat lung apex patients in the future? If so, why? What are the remedies for these rib fractures if the benefits still outweigh the rib fractures?

Specific Comments

              Lines 97-102 of Section 2.4: Can you add more details about the rib fracture toxicities and discuss future options?

Author Response

Thank you for the helpful comment. There were 2patients who had peripheral motor neuropathy, whose tumor were at lung apex. If patients did not received treatment, they died by lung cancer. Moreover, their symptoms were mild. So, we think those were acceptable toxicities. However, we should reduce those toxicities as far as possible. Now, we consider dose constraints for peripheral motor neuropathy.

To respond your comment, we added following sentences in Results and Discussion section.

All of 4 patients who had symptomatic rib fractures relieved by using analgesic. On the other hand, (lines 100-101of page5)

Regarding other toxicities, there were some patients who had grade 2 peripheral motor neuropathy and rib fracture. If patients with lung cancer did not receive radical treatment, the OS time was short [2, 3]. Moreover, symptom of rib fracture can be controlled by analgesic, and symptom of peripheral motor neuropathy were mild. So, we think those toxicities were acceptable. However, we should explain those risk, especially when tumors were at lung apex, and reduce those toxicities as far as possible. To solve those problems, dose constraints for peripheral motor neuropathy is currently under consideration. (lines 183-189 of page 7)

Reviewer 2 Report

This paper proved the efficacy and safety of single fraction CIRT in patients with NSCLC. 

Please answer two questions.

Q1. The tumor located at mediastinal side on Figure 3. However, I think you cannot perform CIRT at the tumor located near the pulmonary hilum. You should mention about that.

Q2. After CIRT, is there a patient who received chemotherapy?

Author Response

Q1. The tumor located at mediastinal side on Figure 3. However, I think you cannot perform CIRT at the tumor located near the pulmonary hilum. You should mention about that.

→ Thank you very much for the great advice. As you say we did not treat for the rumor located near the pulmonary hilum by using single fraction CIRT. To respond your comment, we added following sentences in Methods section.

In present study, there were no NSCLC near the pulmonary hilum, because we treated those lung cancers by CIRT in 12 fractions instead of single fraction. (lines 214-216 of page 8)

Q2. After CIRT, is there a patient who received chemotherapy?

→ Thank you for the helpful comment. To respond your comment, we added following sentences in Results section.

There were no patients who received adjuvant chemotherapy after CIRT. (lines 59-60, page 2)